# Preaching Outside the Temple: On the Literary Witness of James Baldwin as the Word Made Public

## Eric Lewis Williams

The Divinity School, Duke University, Durham, NC 27708-0968, USA; ewilliams@div.duke.edu

**Abstract:** It was the late Bishop Ithiel Conrad Clemmons, former minister of the First Church of God in Christ of Brooklyn, New York, who said of the late famed novelist/essayist James Baldwin that "he was America's inside eye on the Black Holiness and Pentecostal Churches". Though Baldwin admitted that the culture and ethos of the African-American Pentecostal church were "highly significant and indelibly imprinted upon him", according to Baldwin, his faith community's "naiveté about life appalled him and drove him away". While Baldwin left behind the church of his youth, never to return, for the remainder of his writing career, the "backslidden" minister's literary musings continued to be informed (in both style and content) by the formative religious tradition that he left behind. Though several studies have been undertaken that examine Baldwin's significance to various aspects of the study of African-American religion and culture, precious little has been written regarding Baldwin's continuing engagement of the idiom of African-American preaching, the idiom which cultural critic Michael Eric Dyson has nominated as the "jewel in the crown of Black Sacred Rhetoric". While many studies of Baldwin include the fact that Baldwin was a preacher's son and that Baldwin himself preached for a time during his youth, the account is yet to be given of how Baldwin's writings continued to employ the rhythms, grammar, tones, and textures of the Black sacred rhetorical tradition, especially from beyond the borders of the African-American church. This essay seeks to expose not only how Baldwin self-consciously continued to stand in the rhetorical trajectory of the African-American preaching tradition, but the attempt is also to reveal how the writer secularizes the idiom, providing the Black Holiness preacher a hearing from beyond the church. Through a focus on Baldwin as a Black sacred rhetorician, sermonizing from beyond the church, this essay participates in the nearly 100-year-old conversation instigated by the early African-American literary and cultural critic James Weldon Johnson in *God's Trombones: Seven Negro Sermons in Verse* (1927), regarding the neglected significance of the sermon and the preacher in African-American literature and Black expressive cultures. Baldwin's sermonic is here examined as a highly distinctive mode of Black public theologizing.

**Keywords:** black sacred rhetoric; preaching; Baldwin; church; sanctified; salvation; Pentecostal; holiness; soul





"So, in my case, in order to become a moral human being, whatever that may be, I have to hang out with publicans and sinners, whores and junkies, and stay out of the temple where they told us nothing but lies anyway". See (James Baldwin 1971, p. 89).

"They [my childhood friends] still believed in the Lord, but I had quarreled with him and offended him and walked out of his house". See (J. A. Baldwin 1972, p. 16).

## 1. Introduction: Preacher Baldwin on Black Sacred Rhetoric

In a letter penned on the stationary of Atlanta's Ebenezer Baptist Church, dated 26 September 1961, the congregation's young pastor would write to the one-time minister, author, and social critic James Arthur Baldwin. Being deeply impressed by both his

rhetorical giftedness and his highly compelling perspectives on American race relations, Ebenezer's pastor would write:

> "Dear James,
>
> Please excuse me for taking the prerogative to address you by your first name, but I feel just that close to you. I have just finished reading Nobody Knows My Name, and I simply want to thank you for it. This collection of essays and lectures will certainly go down as a classic on the meaning of the social revolution that is taking place in the United States in the area of race relations. It is written with a lucidity of style and a profundity of thought that inspires any serious reader. Your analysis of the problem is always creative and penetrating. Your honesty and courage in telling the truth to white Americans, even if it hurts, is most impressive. I have been tremendously helped by reading the book, and I know that it will serve to broaden my understanding on the whole meaning of our struggle". See (King 1961).

Concluding his letter with great homiletical flourish, the young pastor would

> go on to say,
>
> "You are not only a great Negro writer; you are a great writer. In a most creative way you rose up from all of the crippling restrictions that a negro born in Harlem faces... and plunged against a cloud-filled night of oppression new and blazing stars of inspiration. You make all of us proud to be Negroes.
>
> With warm personal regards....Martin Luther King Jr". See (King 1961).

Knowing of Baldwin's ministerial past within the Black churches of Harlem, one wonders if what King heard in Baldwin's writing was the echo of Baldwin's preacherly voice. Whatever the case, it was clear to King and countless others that Baldwin's literary and homiletical giftings, in increasingly capacious ways, would soon be made known to the world It was the late Bishop Ithiel Conrad Clemmons, former minister of the First Church of God in Christ of Brooklyn, New York, who said of the late famed novelist/essayist James Baldwin, that "he was America's inside eye on the Black Holiness and Pentecostal churches". See (Clemmons 1996, p. vii).

Though Baldwin admitted that the culture and ethos of the Pentecostal church were "highly significant and indelibly imprinted upon him", according to Baldwin, his faith community's "naiveté about life appalled him and drove him away". See (Clemmons 1996, p. vii). While Baldwin disaffiliates from the church of his youth, never to return in the same manner, for the remainder of his literary career, the backslidden preacher's compositions and public addresses continued to be informed in both style and content by the formative religious tradition that he leaves behind. Though several studies have been published that examine Baldwin's significance to various aspects of the study of African-American religion and culture, precious little has been written regarding Baldwin's continuing engagement of the idiom of African-American preaching, the discourse which Michael Eric Dyson has nominated as the "jewel in the crown of Black sacred rhetoric". See (Dyson 1995, p. 16).

While many studies of Baldwin include the fact that he was the son of a preacher and that Baldwin himself, in response to a sense of divine calling, preached for a time during his youth, the account is yet to be given of the ways in which Baldwin's writings continued to employ the rhythms, grammar, tones, and textures of the Black sacred rhetorical tradition, especially from beyond the bounds of the African-American church. This essay seeks to expose not only how Baldwin self-consciously continued to stand in the rhetorical trajectory of the Black preaching tradition, but the attempt is also to reveal how the writer secularizes the idiom, providing the Black preacher a "waiting" congregation beyond the church.

Through a focus on Baldwin as a Black sacred rhetorician, sermonizing from beyond the church, this essay advances the nearly 100-year-old conversation instigated by the early African-American literary and cultural critic James Weldon Johnson in *God's Trombone: Seven Negro Sermons in Verse* (1927), regarding the neglected significance of the sermon and the preacher in African-American literature and Black expressive cultures.

See (Johnson 1927, p. 11). In as much as the sermon can be understood as a repository of Black culture, and the preacher as agent and custodian of this tradition, giving rise to this distinctive, cultural form, when examining Baldwin within this frame, Johnson's observation creates new vistas that emerge for understanding African-American preaching in the secular space. In fact, the Black sermonic form, with its cadence, theological content, hermeneutics, and tonality, became a gift bequeathed to modernity by enslaved Africans. Even shorn of its ecclesial significance, the sermon invites us to examine Baldwin's work, its impact on the wider culture, and the value of these insights and commentary as cultural reservoirs. Baldwin's secular sermonizing provides us with a model and occasion for evidencing Johnson's observation of the neglected significance of the sermon and how it provides elements of importance in understanding contemporary Black expressive culture.

## 2. In the Shadow of the Holy Ghost: Baldwin, Black Religion, and the Study of Black Pentecostalism

Since the publication of his first novel, *Go Tell It on the Mountain* (1953), in which he unveils the chaotic beauty of his formative, Black Pentecostal tradition and exposes its highly percussive spirituality to the world, the writings of James Baldwin have held a treasured standing in both African-American literary and cultural studies. See (J. Baldwin 1953). Due to the expansive scope of his literary, moral, and social vision, Baldwin's canon has been critically engaged by students of theatre, literature, and cultural studies and a wide range of readers from a multiplicity of academic and vocational contexts.

Since its birth as an intellectual trajectory in the late 1960s, "black theology has become and remains the dominant paradigm for the contemporary study of African-American moral and theological thought". See (Sanders 1995, p. 1). With Black theology's penchant for turning toward sources internal to the Black religious experience, African-American literature becomes a critical site of engagement for reflections on and expressions of African-American moral and religious thought. Because of Baldwin's deep rootedness within and seemingly radical rejection of his formative Black Pentecostal tradition, Baldwin's corpus remains an enigmatic trove for students of religious studies, in general, and the African-American religious experience, in particular.

Though several scholars have noted Baldwin's Black Pentecostal moorings and that he was converted within and preached for a time during his youth within the Black Holiness church, few have dared to probe the significance of his engagement of this tradition in their interpretations of the writer. Simply dismissing the essentially Pentecostal accent of his religious experience as "fundamentalist, conservative or evangelical", the attempt of scholars writing from outside of the tradition, aiming at a particular reading of the writer, overlooks both the particularity and peculiarity of the imprint of his formative Pentecostal tradition. These scholars seek to normativize Baldwin's religious experience, thus reducing and simultaneously missing the surplus of religious meaning in many of his insights. While this reductionism suggests the pluriform expression of his practices is common to the whole of Black-Protestant Christianity, any serious attempt to understand Baldwin as both a literary and religious thinker, within and outside of the church, would first have to come to terms with his formative experience within the sanctified church. This is the locus of what Baldwin himself called his "prolonged religious crisis", in which he says he "discovered God, His saints and angels, and His blazing hell". See (J. A. Baldwin 1963, p. 27). Not only has contextualizing Baldwin been problematic for scholars interpreting him from outside of his tradition, but the problem of situating Baldwin has also been a challenge for initiated scholars, those reflecting on Baldwin's corpus from within the Pentecostal and Holiness traditions.

For scholars working from within Holiness and Pentecostal traditions, because of Baldwin's keen insights into and informed critiques of the tradition, the task of reckoning with Baldwin has also proven to be somewhat complex. A close examination of references made regarding Baldwin by scholars from within the tradition suggests that the tendency of Holiness and Pentecostal scholars is to celebrate Baldwin on the one hand and to critique

and distance him from the movement on the other. For James Tinney, founding editor of, once hailed as "the premier corpus of Afro-Pentecostal thought" See (Tinney 1979, p. 29), which began publication in 1977, though willing to extol Baldwin's writings as among the earliest scholarly treatments of Black Pentecostalism, Tinney accuses Baldwin of attempting "to exorcize his haunted memory [of Pentecostalism] through his various novels". See (Tinney 1979, p. 29). Tinney's suspicion of Baldwin is characteristic of the position held by a number of scholars within Holiness and Pentecostal traditions William Turner, a Duke University theologian hailing from the United Holy Church of America, in an article published in 1991, while crediting Baldwin for reflecting "carefully and accurately on the dimension [of Black Pentecostalism] that was of interest to him", dismisses Baldwin's descriptions as "caricatures" and describes the writer's depiction of the tradition as "an essentially negative portrayal". See (W. C. Turner 1995, p. 42).

For theologian Walter Hollenweger, global Pentecostal *doyen*, and professor emeritus of the University of Birmingham, in the United Kingdom, while willing to speak of Baldwin as a "victim of Pentecostalism" "torn between his [Pentecostal] experience and [Pentecostal] theology", See (Hollenweger 2005) Hollenweger further asserts that the writings of Baldwin and others like him show "what damage can be done by people who confuse their religious ideology with the gospel". See (Hollenweger 2005) Bishop Ithiel Conrad Clemmons, a historian and theologian from the Church of God in Christ, while crediting Baldwin as *America's inside eye on the Black Holiness and Pentecostal churches*", says that Baldwin like [other] "Black scholars of [Baldwin's] period" "accept[ed] the white stereotypes of [Black Pentecostal churches as] black sects and cults". See (I. C. Clemmons 1992, p. vii). Black Holiness scholar Cheryl Sanders says of Baldwin that though he was "literally a child of the Sanctified church", he "exiled himself to the margins of a black culture whose richness, beauty, and eschatological hope had transformed and nurtured him". Sanders goes on to say, "[d]espite his efforts to distance himself from the Sanctified church, Baldwin describes with great eloquence its permanent stamp and imprint upon his being[.]". See (C. J. Sanders 1999, p. 112).

And though Baldwin's work has been noted as significant to the study of both Black religion and Black Pentecostalism, there is yet wanting a critical theological treatment of Baldwin that takes seriously the context and theology that shaped and radically informed his religious consciousness. It is out of this religious *milieu* that Baldwin emerges as a major American literary and cultural critic, leaving a corpus of work that would challenge and inspire generations, as the resurgence of his popularity in the academy and broader culture confirms.

### 3. "Every Good-Bye Ain't Gone": Baldwin and the Semblance of Ecclesial Disaffiliation

Though Baldwin's experience in the Sanctified church had given him life in numerous ways, as time evolved, because of the narrowness of his tradition's theological and social vision, Baldwin would soon become utterly disenchanted. In reflecting upon his conscious decision to disengage from his community, Baldwin would later say that he "left the church to save his soul".See (J. A. Baldwin 1972, p. 16). And though Baldwin would admit to having "quarreled with [God] and offended [God] and walked out of [God's] house", See (J. A. Baldwin 1972, p. 16) a closer reading of Baldwin suggests as he himself states with proverbial wit that "every good-bye ain't gone".[1] Moreover, Baldwin would confess it was doubtful that he could ever truly leave the dynamic tradition in which he was nurtured. In the oft-quoted, backsliding scene of his widely acclaimed *Fire Next Time*, says Baldwin, "the church was very exciting. It took a long time for me to disengage from this excitement, *and on the blindest most visceral level, I never really have and I never will*". See (J. A. Baldwin 1963, p. 43). Having left the oppressive and suffocating confines of his local storefront, Pentecostal assembly, Baldwin would now be compelled to share his reconstructed post-Pentecostal vision with the world. This would be accomplished through his writings, theatrical productions, activism, and public speaking engagements. It would

be through his literature and other works that the former minister would curiously remain in conversation with the church. Though Baldwin saw himself as "one of God's creatures whom the church most betrayed", See (J. A. Baldwin 1968, p. 371). by his own admission, Baldwin confesses that he spoke "as one who had always been outside [of the church] even though [he] tried to work in it". See (J. A. Baldwin 1968, p. 371). It was now through his construction of a wider public platform that much of this working inside of the church from the outside would be performed.

### 4. If I Were Still in the Pulpit: Preaching the Word Outside the Temple

In a moment of brutal honesty and existential reflection regarding the person he had become, Baldwin admits, "I hazard that the King James Bible, *the rhetoric of the storefront church, something ironic and violent and perpetually understated in Negro Speech*—and something of Dickens' love for Bravura—Have something to do with me today". See (J. A. Baldwin 1955, p. 5).

It is interesting to note from the aforementioned comment that two of the three sources that the writer admits to having shaped his speech and prose are those that took hold of him in his Pentecostal background. That Baldwin would credit the King James Bible as being a critical fount of his rhetorical influence is not surprising given his former ministerial vocation within the Black Holiness church. That Baldwin continuously engaged scripture is a fact that has not gone unnoticed. However, that Baldwin would credit the rhetoric of the storefront church, something he deemed "ironic and violent and perpetually understated in Negro speech" See (J. A. Baldwin 1955, p. 5). is quite telling indeed as it relates to the peculiar mixture of religious and folk vernacular clearly discernible in his writings. It is in his appropriation of this language that Baldwin's sermonic texture is most clearly made manifest.

Not only is Baldwin's sermonic tenor best revealed in his usage of grammar, but it must also be noted that it is precisely in this way that Baldwin continues to stand in the rhetorical trajectory of his formative Black Church tradition. For though Baldwin has now left the church, it is the protagonists and supporting cast members in his literary productions that are now charged with recovering the sermonic grammar, tones, and textures of the beloved tradition that Baldwin leaves behind. It is now through the mouths of his characters that these sermonic discourses are birthed, giving rise to what might be considered sermonic diversions.

In his widely acclaimed semi-biographical novel, *Go Tell It On The Mountain* (1953), Baldwin delivers vicariously through the mouth of his antagonist a sermonic utterance in which he delineates the costly consequences of the wages of sin. According to his antagonist, Gabriel Grimes, in a moment of unbridled vituperation, the preaching deacon proclaims,

> "it was sin, that drove the son of morning out of Heaven, sin that drove Adam out of Eden, sin that caused Cain to slay his brother, sin that built the tower of Babel, sin that caused the fire to fall on Sodom –sin, from the very foundations of the world, living and breathing in the heart of man, that causes women to bring forth their children in agony and darkness, bows down the backs of men with terrible labor, keeps the empty belly empty, keeps the table bare, sends our children, dressed in rags, out into the whore-houses and dance halls of the world". See (J. Baldwin 1953, p 97).

The cadence, rhythm, allusions to scripture, and references to sin and grace within the construct of Black suffering and freedom embedded in the aforementioned excerpt are hallmarks of the Black sacred rhetorical tradition. In Baldwin's quotation above, one is granted copious reference to numerous Old Testament passages, and one also observes the condemnatory statements proffered regarding the contemporary pleasures of sin, both of which are reminiscent of the tradition from which Baldwin claims to have disengaged. That Baldwin's writings possessed sermonic qualities is a comment that has been made by more than a few. Michael Eric Dyson, in his book *Between God and Gangsta Rap*, maintains that "Baldwin's essays draw equally from the gospel sensibilities and moral trajectory of

the black sermon and the elegant expression of the King James Bible". See (Dyson 1997, p. 126). In the same vein as Dyson, African-American philosopher and cultural critic Cornel West, in his highly provocative text, says of Baldwin that "Baldwin's masterful essays are grounded in moralism, often echoing the rhythm, syncopation, and appeal of an effective Black sermon". See (West 1982, p. 85).West goes on to say, of the explicitly theological nature of Baldwin's writings, that "[t]he salient values of his essays are love, mercy, grace and inner freedom". See (West 1982, p. 85).

Because readers and critics alike all witness the power and effectiveness of Baldwin's pulpitic rhetorical style, it would follow then that just maybe such moments in Baldwin's writings are best understood when read as an extension of his preaching ministry The possibility of reading the sermonic allusions in Baldwin's writings as *literary preaching moments* finds great resonance in a recent homiletical theory advanced by theologian William Clair Turner, professor of Homiletics at the Divinity School of Duke University. Turner has argued that the sermon is most effective when consciously prepared in essayistic form. In an essay entitled, *Preaching the Spirit: The Liberation of Preaching*, Turner argues that the sermon as a theological essay "mediates between the 'friendly-talk' or the 'chit-chat' often heard in the pulpit and the stodgy, incomprehensible treatise that is a mere insertion into the liturgy". See (W. Turner 2005, p. 16). The benefit of this approach for Turner is that "it is bound by the requirement that it be rendered in the language of the people and makes contact with their vernacular". See (W. Turner 2005, p. 16).

To read Baldwin's sermonic diversions as *literary preaching moments* is not inconceivable, seeing that Baldwin himself seemingly conceived of his vocation as a writer and as an extension of his preaching ministry. In a 1976 interview with Jewell Handy Gresham of *Essence* Magazine, Baldwin humorously notes, "[m]y father used to [always] say, 'you didn't call me to preach; God called me to preach.' Many years later I was battling with an editor [about something I had written] and I said [to the editor] "you didn't call me to preach, God called me to preach". See (Gresham 1989, p 160). Baldwin's admission that his literary vocation was a response to God's call to preach sheds great light on the sermonic sensibilities clearly discernible in his essays and other literary productions.

## 5. But the Word of God Is Not Bound: Baldwin, the Sermon and Black Expressive Cultures

In his 1927 publication *God's Trombone: Seven Negro Sermons in Verse*, upon carefully considering the scholarly attention afforded to various folk creations of the American Negro, the early Black literary and cultural critic James Weldon Johnson says of the neglected significance of the folk sermon within American culture, that its significance "has passed unnoticed". See (Johnson 1927, p. 1). Today, nearly 100 years after Johnson's astute observation, the significance of the Black sermon as a genre, given its girth and depth and undeniable imprint upon American culture, remains a topic virtually unexplored. Though hundreds of books have been written regarding the African-American sermon within its ritual and liturgical context, highlighting the role of the preacher and the practice of preaching within the worship life of the Black church, the subject that the sermon has impacted and continues to impact the broader culture, beyond the church, remains a topic of much-needed exploration. As one whose homiletical influence, though incubated within the recesses of the Black Holiness church, bled back into the wider culture through his literary productions, the corpus of James Baldwin becomes a very interesting object of examination for the ways in which his writings bring questions of religion and culture into focus.

That the sermon, a liturgical modality of the church, would sustain such vitality beyond the church is a matter of great theological and cultural significance. Just as Christian musical forms such as hymns, spirituals, and gospel arrangements have demonstrated the ability to cross over into so-called secular arenas and sustain lives of their own, independent of the worshiping community, so it is with the sermonic writings of James Baldwin. While the larger society appropriates Black religious content and tenor in a passive and circuitous

manner, Baldwin actively and unapologetically invokes and conjures the content, form, rhythms, and sonority of Black sacred rhetoric and creates new space, interjecting his work within the broader culture's psyche, deliberation, identity, and expression.

As Baldwin wanders from the confines of the church, a form of self-imposed exile, as Cheryl Sanders intimates, and ventures into the world, he creates a new kind of ecclesial space where souls are reclaimed, and the issues that afflict and infect humanity are laid bare and graced with hope, mercy, and transformative possibility. This is the gift and genius of the Black church tradition that finds an equally gifted writer, James Baldwin, as its evangel. Baldwin constructs a cathedral where no one is excluded from its realm of concern and care. Baldwin decouples notions of value and worth from specific religious, ethnic, and racialized categories and identifies and affirms the sacred worth of all persons. Baldwin locates and affirms the sacred outside the temple and carries the unbound word, unfastened from the moorings of ecclesial tradition and broadcasting it everywhere. Through Baldwin's sermonic writings produced to be accessed by the public, well beyond the church, Baldwin emphatically declares that he, in true holiness form, conceives of and claims that "the world indeed is his parish".[2] From the rostrum of Black sacred rhetoric, having "left the church to save his soul". See (J. A. Baldwin 1972, p. 16).

As he intoned, Baldwin at once critiques the church and the world, as he raises new standards of conduct and illuminates a path toward a new possibility of human community for the world in ways that are accessible, persuasive and hopeful. That Baldwin would continue preaching via his literature, though claiming radical disengagement from the church is a matter that deserves further scholarly attention. The implications of these findings will be significant for students of American religion, cultural studies, Pentecostal studies, and a host of other disciplinary trajectories.

Baldwin's transgressive secularization of the Black sacred rhetorical tradition provides us a window to view his own internal warring with his inherited deposit of faith bequeathed to him in his youth while at the same time reconciling his ecclesial traumas, family dysfunctionality, existential disappointments, sexual identity, and the crushing weight of black life in America in a changing world. This intimate acquaintance with marginalization and estrangement elicits a gravitas that is generative of the finest values and vision of the Black Church and its gift to the world. The beauty of it all is that Baldwin brings these salvific and life-affirming insights from outside the temple.

**Funding:** This research received no external funding.

**Institutional Review Board Statement:** Not applicable.

**Informed Consent Statement:** Not applicable.

**Data Availability Statement:** Not applicable.

**Conflicts of Interest:** The author declares no conflict of interest.

## Notes

[1]   See (J. A. Baldwin 1977). To see Baldwin's penchant for the semblance of disaffiliation, though remaining engaged, please see *Every Goodbye ain't Gone.*

[2]   For examples of the ways in which adherents of Wesleyan and Holiness traditions utilize John Wesley's notion of "the world as my parish", please See (Benge and Benge 2007, p. 77).

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
