# Peer review of "Preaching Outside the Temple: On the Literary Witness of James Baldwin as the Word Made Public"

_religions, doi:10.3390/rel14121547_

Round 1

Reviewer 1 Report

Comments and Suggestions for Authors

This is truly a scholarly piece on James Baldwin and offers a fresh perspective on Baldwin as a "secular preacher" who had a strong formation in the church and then applied the rhetorical flourish, social justice message, and theological interpretations to secular messages he transmitted to the black community in his speaking and communications. This article reinterprets the idea of a "sermon" and also revisits the "black preaching tradition" to reveal Baldwin's singular contributions. The author also blends the religious and secular landscapes of African-Americans, demonstrating the fluidity of experience, therapeutic listening, meditation on messages, and inspiration that moved the black community in different spaces and moments. It is a very powerful and thoroughly intellectually rigorous piece. The engagement with the secondary literature seems thorough enough and the many references to Martin Luther King's work and interactions with Baldwin help situate the narrative and shape the author's argument. It is an extremely strong scholarly article.

Author Response

With respect to the comments and suggestions, I am appreciative for the ways in which the commentators seek to improve the essay and have taken the some of the critiques and made related changes. In addition to making the changes on the corrigenda, as recommended,  Reviewer 1 argues that a person that is not familiar with Baldwin would have little appreciation for the content of this essay and would have difficulty accessing its nuanced argument. This is not necessarily the case, as broad categories referencing race, literature, American culture, religion and history are used, with the assumption that the reader has had some access to Black expressive culture within the United States within the last 100 years. Reviewer 1 also asked for additional samples of Baldwin’s performance of preaching in the public space.   Given the amount of examples previously provided in the text, I felt there were ample references used in support of the claims I made and felt additional quotes would be excessive.  In a longer project on the subject, more examples would be warranted, but given the scope of this essay, I felt the examples provided more than sufficed.

Reviewer 2 Report

Comments and Suggestions for Authors

The topic of this article shows promise. I was expecting much more work with primary sources from the statement in the abstract from lines 14 through 22.

Also lines 97-100 leads the reader to expect that there will be some analysis of how Baldwin secularizes the idiom.

It would help if there was a clear presentation about what black sacred tradition is earlier in the article.

How is secularism to be defined, too? It would help to know what is considered secular in Black's time of preaching. 

It would be good to see more explanation of the quotation on lines 243 through 250. Explain how this fits with black sacred rhetoric tradition.  Also, how does Black secularize it. Provide your own analysis.

A few other quotes from his preaching with analysis would help, too.

Author Response

With respect to the comments and suggestions, I am appreciative for the ways in which the commentators seek to improve the essay and have taken the some of the critiques and made related changes. 

With respect to Reviewer 2, and the request to expand on secularism, his introduction of the notion of secularism is foreign to the goals of this paper and was never here invoked.  Secularism as a dogmatic posturing and ideology are never invoked. The term ‘secular’ is used here to refer to Baldwin’s employment of black preaching outside the temple, or the ritual worship context wherein African american preaching is performed.

Furthermore, Reviewer 2 asks for an explanation of how a quotation fits within the black sacred rhetorical tradition.  Such explanation was provided and further expanded in the section immediately following the referenced quotation.

Reviewer 3 Report

Comments and Suggestions for Authors

This essay has great potential. As it stands, it is excellent for those familiar with writings of James Baldwin. However, general readers and others interested on the impact of evangelicalism and Pentecostalism in Black studies would want more. Perhaps this search for more is the merit of the essay.

Within the circle of students and scholars of Baldwin, this could provoke interesting conversation. But for people with general or little knowledge of Baldwin, this essay needs to be expanded

Could the author, for example, do exactly what author suggests at the conclusion: "That Baldwin would continue preaching via his literature though claiming radical disengagement from the church is a matter that deserves further scholarly attention."  More samples of this performance for the general reader?

CORRIGENDA:

Line 266 - Sentence is incomplete

Line 269: Turner not "Tuner"

Author Response

With respect to the comments and suggestions, I am appreciative for the ways in which the commentators seek to improve the essay and have taken the some of the critiques and made related changes.

With respect to Reviewer 3, I was greatly encouraged by the review and the reviewer’s grasp of the goals and intent of the essay, not as a conclusive and definitive word on the subject; rather  one that explores the future direction for consideration of Baldwin’s work and the contexts from which it emanates.

Round 2

Reviewer 2 Report

Comments and Suggestions for Authors

one space after footnote on lines 81, 92, 105, 122, 149, 170, 201, 222. Please check other spacing after footnote numbers.

See spacing on line 157 after word "references" and lines 162 and 163 "of Pentecostalism", line 167 "published in", line 178 "period"   "accepting" and 179 "cults.  footnote number 17

line 196 has too many spaces between period and next sentence.

font size in footnote 35

Author Response

Thank you for your comments. I have made the requested edits